# Nuclear Receptor Atlases of Choroidal Tissues Reveal Candidate Receptors Associated with Age-Related Macular Degeneration

**DOI:** 10.3390/cells11152386

**Published:** 2022-08-02

**Authors:** Jeremy Peavey, Vipul M. Parmar, Goldis Malek

**Affiliations:** 1Duke Eye Center, Department of Ophthalmology, Duke University School of Medicine, Durham, NC 27710, USA; jeremypeavey@gmail.com (J.P.); vipulkumar.parmar@duke.edu (V.M.P.); 2Department of Pathology, Duke University School of Medicine, Durham, NC 27710, USA

**Keywords:** choroidal endothelial cells, nuclear receptor atlas, choroidal injury, age-related macular degeneration

## Abstract

The choroid is a vulnerable tissue site in the eye, impacted in several blinding diseases including age related macular degeneration (AMD), which is the leading cause of central vision loss in the aging population. Choroidal thinning and choriocapillary dropout are features of the early form of AMD, and endothelial dysfunction and vascular changes are primary characteristics of the neovascular clinical sub-type of AMD. Given the importance, the choroidal endothelium and outer vasculature play in supporting visual function, a better understanding of baseline choroidal signaling pathways engaged in tissue and cellular homeostasis is needed. Nuclear receptors are a large family of transcription factors responsible for maintaining various cellular processes during development, aging and disease. Herein we developed a comprehensive nuclear receptor atlas of human choroidal endothelial cells and freshly isolated choroidal tissue by examining the expression levels of all members of this transcription family using quantitative real time PCR. Given the close relationship between the choroid and retinal pigment epithelium (RPE), this data was cross-referenced with the expression profile of nuclear receptors in human RPE cells, to discover potential overlap versus cell-specific nuclear receptor expression. Finally, to identify candidate receptors that may participate in the pathobiology of AMD, we cataloged nuclear receptor expression in a murine model of wet AMD, from which we discovered a subset of nuclear receptors differentially regulated following neovascularization. Overall, these databases serve as useful resources establishing the influence of nuclear receptor signaling pathways on the outer vascular tissue of the eye, while providing a list of receptors, for more focused investigations in the future, to determine their suitability as potential therapeutic targets for diseases, in which the choroid is affected.

## 1. Introduction

The human choroidal endothelium is a highly fenestrated vasculature responsible for supplying blood to the posterior pole of the eye [1]. This tissue spans the ora serrata to the optic nerve and is adjacent to Bruch’s membrane and the retinal pigment epithelium (RPE), nurse cells to the neurosensory retina [2]. Morphologically, the choroid can be broadly divided into three vascular layers (from the inner to the outer layer): the choriocapillaris (composed of small fenestrated vessels), Sattler’s layer (containing medium-sized vessels), and Haller’s layer (containing larger vessels) [3]. Furthermore, interspersed within the choroidal tissue are micro- and macro-endothelial cells, melanocytes, fibroblasts, and resident immunocompetent cells [1]. Functionally, the outer blood–retina barrier, formed by the choroid and RPE, regulates the passage of solutes and nutrients into the sub-retinal space [4]. Furthermore, the choriocapillaris and RPE both secrete growth factors and immunosuppressive factors that aid in regulating cell function and contribute to immune privilege within the subretinal space [5]. While the RPE is often considered the primary relevant tissue in the progression of blinding diseases such as age-related macular degeneration (AMD), recent advances indicate that changes in choroidal morphology and function may play a larger role in the development and progression of diseases of the posterior pole, than previously believed [6,7,8].

AMD is a leading cause of blindness and severe visual impairment in the elderly populations of developed countries [9,10], impacting approximately 30% of aged individuals [11,12]. Identification of effective therapies has been challenging, to some extent due to the multivariable etiology of AMD, with genetics, systemic health, and environmental factors contributing to disease development and progression [13,14]. These factors collectively impact a variety of pathways including but not limited to lipid regulation, proteostasis, inflammation, response to oxidative stress, mitochondrial function, and extracellular matrix regulation [15,16,17,18,19,20]. The pathogenesis of AMD is complex on the cellular level and includes progressive dysfunction and degeneration of retinal cells including the RPE, photoreceptors, and choroidal endothelial cells, and cells involved in activation of the immune response, such as resident macrophages and microglial cells [21,22]. Clinically, the early and late dry stages of AMD are characterized by the accumulation of lipid- and protein-rich extracellular deposits below the RPE layer within the macular regions, RPE atrophy, as well as choriocapillary dropout [6,23,24]. In the neovascular subtype of AMD, choroidal vascular changes are also evident, but in contrast to dry AMD, involve abnormal extravasation of the choroidal vasculature, leakage of blood, fluid and/or exudates within the sub-RPE and sub-retinal spaces [8,25]. Importantly, all clinical sub-types of AMD involve choroidal abnormalities, despite some differences in their presentations. Given the vital role of the choroid in physiologically supporting the RPE and neural retina, elucidating the biologic underpinnings of this tissue type is an important first step in the goal to identifying signaling pathways that may serve as potential therapeutic targets.

Nuclear receptors (NRs) are the largest superfamily of ligand-dependent transcription factors in the human genome. Functionally, they regulate a variety of cell homeostatic events and as such are important in many aspects of human development and disease [18,26]. Currently, there are a number of FDA-approved drugs on the market that target these receptors, accounting for approximately 16% of small molecule drugs [27], including several directed to systemic and neurodegenerative diseases that share common pathogenic pathways with AMD [28,29,30]. This precipitated the initiative to conduct a comprehensive evaluation of the expression of nuclear receptors in the eye under normal and disease conditions. Previously, we created a nuclear receptor atlas of human RPE cells, given their vulnerability in AMD [31]. We determined the expression of all 48 human nuclear receptors in three cellular models systems: ARPE19 cells, a spontaneously arising RPE cell line derived from a 19 year old donor; primary human RPE cell cultures, established from RPE cells harvested from human donor eyes; and freshly isolated RPE cells, not subjected to culture conditions [31]. The results of that study helped establish the eye as a potential secondary endocrine organ and identified several candidate receptors important in RPE biology and disease [17,32,33,34,35]. Herein, we continue investigating nuclear receptor biology in the aged eye with a focus on the outer retinal vasculature using two model systems: primary human choroidal endothelial cells and freshly isolated choroidal tissues from human donor eyes. As cross-talk between RPE cells and the choroid, in vivo, is an important aspect of the ocular biology of the posterior eye cup, we compared the two atlases and categorized potential nuclear receptors as important in RPE-choroid biology. Finally, we profiled nuclear receptor expression in the choroid under disease conditions using a pre-clinical experimental mouse model of wet AMD, to identify candidate nuclear receptors important in outer retinal neovascular lesion development.

## 2. Materials and Methods

### 2.1. Model Systems

Primary choroidal endothelial cell lines (1°CECs) and freshly isolated choroidal (fCh) tissue were isolated from adult donors (1°CECs, *n* = 8, 2 females and 6 males, ages 47–90 years; Appendix A; fCh, *n* = 6, 3 males and 3 females, ages 61–90 years; Appendix A). Human donor eyes were collected by BioSight, NC, USA and processed within 6 h from time of death for freshly isolated tissue use and within 8 h from time of death for primary CEC culturing. Donor eyes used in this study did not have a past history of ocular disease. For freshly isolated choroid, donor eyes were sliced 1–2 cm at the equator and stored at 4 °C in RNAlater (Invitrogen; Waltham, MA, USA) for no more than 48 h before choroidal tissue isolation. For dissection, the anterior segment, vitreous, and neural retina were carefully excised. RPE cells were carefully scraped away with a surgical blade without breaking through Bruch’s membrane and the cup was washed with PBS (Gibco; Paisley, UK). Sections of choroid clear of observable RPE debris were cut and placed in a clean Petri dish and kept moist with RNAlater. Choroidal tissue was diced into pieces approximately 0.5–2 mm in size. Choroidal pieces were maintained in RNAlater until RNA isolation and cDNA preparation.

For primary CEC culturing, donor eyes were obtained and anterior segment, vitreous, neural retina, and RPE were removed as described above. Choroidal tissue was dissected into small pieces and digested in 0.2% collagenase type 1A (Sigma-Aldrich; C2674, St. Louis, MO, USA) in a shaking water bath for 30 min at 37 °C or until single cell suspension was achieved and neutralized with 10% FBS EGM-2MV endothelial cells media (Lonza; CC-3202, Walkersville, MD, USA) supplemented with 1× non-essential amino acids (Corning; 25-025-CI, Manassas, VA, USA), 1× Penicillin-Streptomycin (Sigma-Aldrich; P0781), and 2.50 μg/mL amphotericin B (Gibco; 15290018, Burlington, ON, Canada). Cells were passed through a 40 μM separation filter (Corning; 352340, Durham, NC, USA), rinsed in PBS, and centrifuged for 5 min at 250× *g* (10 °C). Cells were resuspended in EGM-2MV, stained with Human CD144 Microbeads (Miltenyi; 130-097-857, Auburn, CA, USA), and blocked with FCR-blocking reagent (Miltenyi; 130-059-901). CECs were enriched via magnetic-activated cell sorting on a LS midiMACS separator (Miltenyi; 130-042-301). LS columns were washed five times and eluted with EGM-2MV media. CECs were plated on 5 μg/mL human fibronectin/0.5% gelatin (Sigma-Aldrich; F0895, Sigma-Aldrich; G1393, St. Louis, MO, USA) coated 6-well plates and grown to confluence. At 90% confluence, CECs were dissociated into a single-cell solution with Accumax (Invitrogen; 00-4666-56, Carlsbad, CA, USA) and labeled with CD31-PacificBlue conjugated antibody (Biolegend; 303114, San Diego, CA, USA) and Aquablue Live-Dead Stain (Invitrogen; 43110, Eugene, OR, USA). Fluorescent-activated cell sorting (FACS) was used to establish a pure population of choroidal endothelial cells, which were validated and characterized. Penicillin-Streptomycin and amphotericin B were removed from the media at passage 2 and cells up to passage 5 were used.

### 2.2. Tube Formation Assay

Tube formation assay was used as an in vitro angiogenesis model to validate 1°CEC function as previously described [36]. Briefly, Geltrex™ (Life Technologies, Grand Island, NY, USA) was thawed overnight at 4 °C. Using cold pipette tips, 10 μL/well of Geltrex™ was added to a μ-slide angiogenesis plate (ibidi GmBH, Gräfelfing, Germany). The Geltrex™ solidified into a thin layer after incubation at 37 °C for 1 h. 1°CECs were plated at a density of 12,000 cells/well and incubated at 37 °C, in 5% CO_2_. After 3 h, network formation was imaged using an inverted phase-contrast microscope (Zeiss, Axio Observerer.D1, Whiteplains, NY, USA).

### 2.3. RNA Isolation and cDNA Preparation

Total RNA was isolated from 6 samples of freshly isolated choroidal tissue and 3 consecutive passages of all 8 donor CECs at confluence as described previously [31,32]. The RNeasy kit (Qiagen, Valencia, CA, USA) was used to extract total RNA with DNAse I treatment to eliminate gDNA contamination. RNA quality was measured with nanodrop spectrophotometer (DeNovix DS-11 spectrohpotometer, Wilmington, DE, USA) and an Agilent Bioanalyzer (Santa Clara, CA, USA). cDNA synthesis from 1 µg of RNA was done using iScript™ cDNA Synthesis Kit (Biorad, Hercules, CA, USA).

### 2.4. AMD-Specific Genotyping

cDNA from donor tissues or cells were amplified and used for sequencing. Regions containing SNPs of CFH, CFI, C2, C3, CFB, ARMS2, HTRA1, APOE were amplified by PCR and sequenced at GENEWIZ Sanger Sequencing. The genotype of the donor tissue used to establish cell culture lines are listed in Appendix A.

### 2.5. Primer Validation, qPCR Assay

List of all receptors evaluated including the gene symbol, unified nomenclature and common names are provided in Appendix A. Primers for the receptors and housekeeping genes were selected from the NURSA website and from Harvard primer bank and are listed in Appendix A. Primer oligonucleotides were purchased from Integrated DNA Technologies at 25 nM scale with standard desalting. Primers were validated with the observation of a single peak in the dissociation curve. iQ SYBRGreen Supermix (Bio-Rad) was used to prepare PCR reactions, 25 ng cDNA was used per reaction and samples were run on a CFX iCycler (Bio-Rad iCycler CFX96 Thermo1000). qPCR was used to determine the nuclear receptor expression levels in both model systems. *ALAS-1*, a mitochondrial enzyme, was used as a housekeeping gene based on its low variability and high stability in endothelial cells [37]. Gene expression was calculated by using the relative absolute standard curve method, in which all the sample Cts are calculated from the reference cDNA standard curve and then normalized to the housekeeping gene [31]. qPCR assays were run in 96-well plates with triplicate experimental samples of a standard curve of the human reference cDNA and triplicates of three biological replicates of the 1°CEC and fCh. To avoid genomic amplification, control samples included no reverse transcriptase control and to rule out formation of primer dimers, no template control was included. Nuclear receptors for which the Ct values were greater than 35 in two or more biological replicates per cell type were considered absent (Appendix A).

### 2.6. Venn Diagrams and Pathway Analysis

Venn diagrams were created using the online software VENNY2.1 (http://bioinfogp.cnb.csic.es/tools/venny/index.html) created by Juan Carlost Oliveros (Madrid, Spain; software was accessed on 2 September 2021). All gene sets were submitted to Venny 2.1.0 as different categories, and the output Venn diagram is presented. Panther analysis was used for Gene Ontology analysis where biological process refers to the broader, overarching processes that the gene product helps to accomplish. Gene sets of nuclear receptors were analyzed using the online functional anotational tool DAVID Bioinformatics resources (Bethesda, MD, USA; https://david.ncifcrf.gov/) [38,39]. Software was accessed on 25 April 2022 with Reactome database (https://reactome.org/) [40]. Software was accessed on 25 April 2022.

### 2.7. Experimental Laser Induced Choroidal Injury in Mice

Fourteen C57BL/6J mice (24–28 months; 3 female, 11 males) subjected to experimental laser induced injury served in our animal model group and 13 mice (24–28 months; 2 female, 11 males) not subjected to laser injury served in the control group. Mice were randomly assigned to each cohort. The model of CNV in mice was established as described previously [32,34,36,41] and below. Mice were dilated with a tropicamide/phenylephrine mix (1:1 of 1% Tropicamide, Cat# NDC0404-7192-01, Henry Schein Inc., Melville, NY, USA and 2.5% Phenylephrine-HCl, Paragon Bioteck Inc., Portland, OR, USA, Cat# NDC42702-102-15). Fixing the mice in front of the slit lamp with fully open upper and lower eyelids, 12 thermal burns were induced by a 650 nm diode red laser (at a 75 μm spot size, 150 mW intensity, and 100 ms duration) at approximately 1 pupillary distance from the optic disc in both eyes of mice. Eye ointment (Gonak Hypromellose, 2.5%, Cat# NDC17478-064-12, Akorn Animal Health Inc., Lake Forest, IL, USA) was applied to the eyes and mice were given time to recover. The mice in the control group were fed normal chow and not subjected to laser treatment. Fluorescence leakage of choroidal lesions in mice was observed by fluorescein angiography after three days. Mouse eyes were harvested four days post-CNV laser treatment. Following enucleation, mouse choroidal tissues were isolated by making a cut around the ora serrata and cornea, removal of lens and vitreous body. Next the retina was gently peeled off avoiding disturbing the RPE monolayer. The RPE was removed using a paint brush and the choroid was scraped from the sclera and stored in TRIzol reagent at −80 °C until RNA extraction. 

### 2.8. In Vivo Imaging

A Micron IV (Phoenix Research Laboratories Inc., Pleasanton, CA, USA) retinal imaging microscope was used for in vivo imaging. Fundus images were obtained from anesthetized C57/BL6J mice. The optical coherence tomography (OCT) module of the Micron IV was used to image retinal layers guided from the bright field. Fluorescein angiography was also obtained on anesthetized mice following an intraperitoneal injection with 10% sodium fluorescein (AK-FLUOR, Akorn, Decatur, IL, USA) at a dose of 0.02 mL/5–6 g body weight [34].

### 2.9. RPE-Choroid Flat Mount Preparation

Whole RPE-choroid flat mounts were prepared to confirm CNV lesion formation. Isolectin GS-IB_4_ staining (Invitrogen; I21411, Carlsbad, CA, USA) allowed visualization of vascularity and propidium iodide (Sigma-Aldrich; P4170, St. Louis, MO, USA) was used to evaluate cellularity of the neovascular lesion [32,36]. Lesions were evaluated under a Nikon C2si confocal microscope. Horizontal optical sections or z-stack images of the flat mount were obtained at 1.50 μm intervals using NIS-elements microscope imaging software (Nikon, Melville, NY, USA)). Images of the entire flatmount were captured from 4–5 fields at 4X magnification and stitched together using the Photomerge function of Adobe Photoshop 22.4.2 (Adobe, San Jose, CA, USA).

### 2.10. RT^2^ Profiler PCR Array for Mouse Nuclear Receptors and Coregulators

RNA for PCR array was isolated using Qiagen RNeasy Universal kits (Qiagen; 73404, Hilden, Germany) and cDNA was synthesized using RT^2^ First Strand Kits (Qiagen, MD, USA). RT^2^ Profiler PCR Array for Mouse Nuclear Receptors & Coregulators (Cat. No. 330231 PAMM-056YA, MD, USA) contains a list of 84 pathway-focused genes including chromatin modifying enzymes, as well as five housekeeping reference genes. In addition, each array contains a panel of proprietary controls to monitor genomic DNA contamination (GDC) as well as the first strand synthesis (RTC) and real-time PCR efficiency (PPC). The arrays were run on a Biorad CFX96 real time PCR machine (Hercules, CA, USA), and included 4 biological samples in each group. Each PCR was repeated three times. Data from the PCR array were analyzed using the Geneglobe analysis web portal on the Qiagen website. Fold change was calculated as the ratio of normalized gene expression in CNV-laser sample to normalized gene expression in control group. Specifically, the ΔCT was calculated between gene of interest (GOI) and an average of reference genes (HKG), followed by Δ − ΔCT calculations [ΔCT (CNV Group) − ΔCT (Control Group)]. Fold Change was then calculated using 2 ^(−ΔΔCT)^ formula. The *p*-values were calculated based on a Student’s *t*-test of the replicate 2 ^(−ΔCT)^ values for each gene in the control group and treatment groups. The *p*-value calculation used was based on parametric, unpaired, two-sample equal variance, two-tailed distribution. Fold change reflects fold regulation as it represents fold change in a biologically meaningful way, with a fold change of one or more representing positive up-regulation where fold change is equal to fold-regulation. While fold change values less than one indicate negative down-regulation where fold regulation was calculated as negative inverse of the fold change.

### 2.11. Statistical Analyses and Rigor

Graph preparation and statistical analyses were performed using GraphPad Prism 9 (GraphPad Software Inc. San Diego, CA, USA). The differences between two groups were compared using unpaired *t*-test and differences amongst three or more groups were compared using One-Way ANOVA. More than 2 fold-regulation was considered significant when comparing Control vs. CNV-lasered mice. The *p*-values are calculated based on a Student’s *t*-test of the replicate 2 ^(−ΔCT)^ values for each gene in the control group and treatment groups. The *p*-value calculation used is based on parametric, unpaired, two-sample equal variance, two-tailed distribution. Biological and technical replicates of 3 or more were used in each experiment.

### 2.12. Institutional Review Board Statement

Use of human donor eyes for research was approved by the Duke University Institutional Review Board. Additionally, the Duke University Institutional Animal Care and Use Committee approved the study protocol and all animal experiments were performed in accordance with the guidelines of the ARVO Statement for the Use of Animals in Ophthalmic and Vision Research.

## 3. Results

### 3.1. Rigor of Human Choroidal Endothelial Cell and Human Choroid Tissue Sample Preparation

Two choroidal samples were used in this study: (1) primary cell cultures of human choroidal endothelial cells (1°CECs) harvested and enriched from human donor tissue, and (2) freshly isolated choroid samples (fCh) isolated from human donor eyes. All human donor eyes were collected within a short post-mortem time (Appendix A). Cultures of primary CEC were used to determine the nuclear receptor landscape of endothelial cells, while the freshly isolated choroid was investigated to reveal the in situ expression of nuclear receptors in the entire outer vascular tissue, whilst avoiding the possibility of acclimatization of cells to culture conditions or de-differentiation. As there is heterogeneity amongst humans, we analyzed 8 primary cell samples harvested from donors aged 47–90 years old and 6 choroidal tissue samples isolated from donors aged 61–90 years old. 1°CECs collected following positive enrichment by CD31 using FACS (Figure 1A) formed a typical cobblestone-like appearance (Figure 1B) and the endothelial surface markers, CD31 and Von Willebrand factor (vWF) were localized using immunohistochemical methods (Figure 1C). 1°CECs were propagated and validated further by a tube-formation assay angiogenesis model, displaying the formation of an intricate network of tubules (Figure 1D).

### 3.2. Nuclear Receptor Atlas of Primary Choroidal Endothelial Cells and Freshly Isolated Choroid

We subjected 1°CECs and fCh to expression profiling, using quantitative real-time PCR, creating the nuclear receptor atlas. The atlas included assessments of the expression of the 48 members of the nuclear receptor superfamily, two isoform variants of the peroxisome proliferator activator receptor beta/delta and gamma (*PPARβ/δ: NR1C2* and *PPARγ: NR1C3*), the aryl hydrocarbon receptor (*AhR*) and its obligate binding partner the aryl hydrocarbon receptor nuclear translocator (*ARNT*). In Figure 2 the expression levels of the nuclear receptors are shown in detail, with the receptors sub-grouped into the typical five classes classified based on known ligand properties. The groups include steroid hormone receptors (Figure 2A); non-steroid hormone receptors, (Figure 2B); adopted orphan nuclear receptors (Figure 2C); and orphan nuclear receptors (Figure 2E). The expression profile of *AhR* and its receptor *ARNT*, are also shown (Figure 2D). Analysis of the expression profiles from 1°CECs revealed the presence of 35 nuclear receptors, of which 10 were expressed at high levels, 12 at medium levels, and 13 at low levels (Figure 2F). Analysis of fCh data showed the expression of 42 nuclear receptors, of which 13 were expressed at high levels, 11 at medium levels, and 18 at low levels (Figure 2G). A total of 17 receptors in 1°CECs and 10 in fCh were considered not detectable due to their high Ct values (Ct ≥ 35).

The expression of *AhR* and *ARNT*, transcription factors that share similar mechanisms of action to nuclear receptors were also evaluated and found to be present at high levels in both model systems. This was predicted as both *AhR* and *ARNT* have been shown to play important roles in vascular biology, angiogenesis, inflammation, and AMD [36,42]. In examining the expression of the receptors in further detail, all isoforms of the retinoic acid receptors *RARα* (*NR1B1)*, *RARβ* (*NR1B2)* and *RARγ* (*NR1B3*), retinoid X receptor alpha (*RXRα*: *NR2B1*), retinoid related orphan nuclear receptors *RORα* (*NR1F1*) and *RORβ* (*NR1F2*) were measured at medium or high levels in both model systems, while *RXRβ* (*NR2B2*) and *RXRγ* (*NR2B3*) were only detectable in fCh. Collectively, these receptors may be critical to retinol transport from the choroidal blood supply to the RPE in support of the visual cycle. The expression of *RORγ* (*NR1F3*) was below the detection level in both cell models. Adopted orphan nuclear receptors including the liver X receptors (*LXRα*: *NR1H3* and *LXRβ*: *NR1H2*), thyroid hormone receptor (*TRα: NR1A1* and *TRβ*: *NR1A2*) and *PPAR**γ1* (*NR1C3*), were also expressed in both model systems. Members of the steroid hormone receptor including the vitamin D receptor (*VDR: NR1I1*), glucocorticoid receptor (*GR*: *NR3C1*), mineralocorticoid receptor (*MR*: *NR3C2*) and androgen receptor (*AR: NR3C4*) were expressed in both model system, while farnesoid X receptor (*FXR*: *NR1H4*) and progesterone receptor (*PR*: *NR3C3*) were only found in fCh. Estrogen receptor alpha (*ERα*: *NR3A1*) was found to be expressed at low levels in both fCh and 1°CEC, while *ERβ* (*NR3A2*) was found exclusively in 1°CECs. Of the estrogen related receptors (*ERR*), the alpha isoform (*ERRα*: *NR3B1*) was found to be expressed in both 1°CECs and fCh while the *ERRβ* (*NR3B2*) and *ERRγ* (*NR3B3*) were absent from 1°CECs. Of the orphan nuclear receptors, while germ cell nuclear factor [*GCNF*: *NR6A1*, also known as retinoid receptor-related testis-associated receptor (*RTR*)] and liver receptor homolog 1 (*LRH1*: *NR5A2*) were expressed in both model systems, its paralog the steroidogenic factor 1 (*SF1*: *NR5A1*) was not detectable in 1°CECs and fCh. Amongst the orphan nuclear receptors, the testicular receptor 4 (*TR4*: *NR2C2*) was expressed in both model systems, while the testicular receptor 2 (*TR2*: *NR2C1*) was only detected in 1°CEC. The three members of the *NR4A* family of nuclear receptors, which are early immediate-response genes, including nerve growth factor IB nuclear receptor variant 1 (*NUR77*: *NR4A1*), a human homolog of nur-related protein-1 (*NURR1*: *NR4A2*) and neuron-derived orphan receptor-1 (*NOR1: NR4A3*) were expressed in fCh. Under culture conditions, however, 1°CECs only retained expression of *NURR1* and *NOR1*, but not *NUR77*: *NR4A1*. The orphan nuclear receptors hepatocyte nuclear factor 4 isoforms *HNF4α* (*NR2A1*) and *HNF4γ* (*NR2A2*) and *TLX* (homologue of the Drosophila tailless gene also known as *NR2E1*) were exclusively expressed in fChs. Interestingly, the photoreceptor-specific nuclear receptor (*PNR*: *NR2E3*) was also found to be present in 1°CEC. Other nuclear receptors classified as absent from both model systems included the constitutive active receptor (*CAR*: *NR1I3*), DSS-AHC critical region on the X chromosome protein 1 (*DAX-1*: *NR0B1*), *PPARγ2* (*NR1C3γ2*), pregnane X receptor (*PXR*: *NR1I2*), and small heterodimer partner (*SHP*: *NR0B2*). Overall, fCh lacked expression of 10 nuclear receptors as opposed to 17 for 1°CECs.

Taken together, these results demonstrate that the expression of the majority of the nuclear receptors overlapped between the freshly isolated choroid and cultured choroidal endothelial cells, supporting the use of primary cultures of choroidal endothelial cells as a platform for further mechanistic studies into nuclear receptor function. Only seven nuclear receptors were absent from 1°CECs as compared to fCh, highlighting possible differences between cells exposed to culture conditions and the in vivo state. On the flip side, only three receptors were absent from fCh as compared to 1°CECs suggesting potential contributions in expression from other populations of cells present in the complex choroidal tissue.

### 3.3. Pathway Analysis of Nuclear Receptors Expressed in Choroidal Endothelial Cells and Freshly Isolated Choroid

Analyses of the atlases revealed 32 common nuclear receptors between 1°CEC and fCh, 3 exclusively expressed in 1°CEC and 10 present only in fChs (Figure 3A). Receptors expressed only in fChs included *ERRβ*, *ERRγ*, *FXR*, *HNF4α*, *HNF4γ*, *NUR77*, *PR*, *TLX*, *RXRβ*, and *RXRγ*, while 1°CEC only expressed *ERβ*, *TR2* and *PNR*. Gene ontology analysis of 32 nuclear receptors common to 1°CEC and fCh revealed a number of biological pathways of which the top 50 are presented, including but not limited to regulation of cholesterol uptake, mitochondrial biogenesis, gluconeogenesis, and plasma lipoprotein clearance (Figure 3B). Importantly, transcriptional regulation by nuclear receptors is heavily demonstrated in the pathway analysis, explicable as nuclear receptors are the largest class of transcription factors in the human genome. General transcription mediated by RNA polymerase II for protein coding genes and gene expression of ribosomal RNA by RNA polymerase I, are also activated by the subset. SUMOylation of intracellular proteins including many nuclear receptors and their target proteins leads to transcriptional repression through various mechanisms and is also signified in the analysis. SUMO E3 ligases that mediate SUMO conjugation to target proteins and other post translational modifications are also enriched in the nuclear receptor subset. Other notable pathways detected are protein metabolism including synthesis, post-translational modifications and degradation; lipid metabolism, an important source of energy and precursor for hormones and vitamin D, also able to serve as activating ligands for nuclear receptors; cellular response to stress; and heme signaling important in the process of scavenging heme in the plasma by plasma proteins making it readily bioavailable. Circadian rhythm and circadian clock related pathways are also represented by *RORα*, *REV-ERBα*, *GR*, *PPARα*, and *RXRα*. Transcription of the master regulator of circadian rhythm *BMAL1* (*ARNTL*) gene is controlled by *RORα* and *REV-ERBα*, both of which not only are targets of *BMAL1*:*CLOCK/NPAS2* in mice, but also compete for the same response element (RORE) in the *BMAL1* promoter [43]. *LXRα* and *LXRβ*, two receptors critical to lipolysis, lipogenesis, cholesterol uptake, bile acid homeostasis, plasma lipoprotein clearance and gluconeogenesis were also highly denoted in the pathway analysis. *AhR* and *ARNT* involvement was epitomized by aryl hydrocarbon receptor signaling and xenobiotics terms. A subset of nuclear receptors was differentially expressed between the two cell models and are listed in Table 1.

### 3.4. Comparison of Choroidal Endothelial Cell and RPE Atlases

The RPE and choroid are an epithelial—vascular complex that works in tandem to support the neurosensory retina, facilitating the supply of oxygenated blood, recycling of retinoids vital for the visual cycle, clearance and renewal of photoreceptor outer segments and provide metabolic support to photoreceptor cells [44]. As such, RPE and choroid dysfunction have been implicated in a number of ocular diseases that result in vision impairment and loss [45,46]. Here, in we cross-compared the results of our 1°RPE and freshly isolated RPE (fRPE) nuclear receptor atlas [31] to that of the 1°CEC and fCh, with the goal of identifying candidate receptors that may play a role in RPE-choroid homeostasis (Figure 4).

Twenty-six nuclear receptors were found to be expressed in all four model systems. These included *LXRα* and *LXRβ*, important regulators of lipid metabolism and inflammation [26,47]; *PPARα* and *PPARβ/δ*, important in bioenergetics and metabolism processes; thyroid hormone receptor nuclear receptors *TRα* and *TRβ*; all three isoforms of RARs (*RARα*, *RARβ* and *RARγ*); the orphan nuclear receptor *RORα* and its paralog *RORβ*; V-ErbA-Related Protein 2 (*EAR2: NR2F6*); V-ErbA-Related Protein 1 (*REV-ERBα*: *NR1D1*) and V-ErbA-Related Protein 1-Related (*REV-ERBβ*: *NR1D2*). Of the ERRs, only *ERRα* was expressed in all four model systems, while *ERRγ* was only expressed in freshly isolated RPE and choroid and *ERRβ* was exclusively expressed in fCh. Germ cell nuclear factor (Retinoid Receptor-Related Testis-Specific Receptor, *GCNF: NR6A1*), which has been shown to be involved in neurogenesis and germ cell development [48] plays a role in all four models. Out of the RXRs, which are capable of heterodimerizing with over 17 other nuclear receptors [49,50], *RXRα* (*NR2B1*) was ubiquitously expressed in the RPE and choroid model systems. However, *RXRβ* expression was not found in 1°CEC and *RXRγ* was not expressed in 1° RPE or 1°CECs. Amongst the nerve growth factor IB-like nuclear receptors, *NURR1* was found to be expressed in all the samples, while *NOR1* expression is lost with culturing of 1°RPE and *NUR77* expression is undetectable in 1°CEC. Expression of steroid hormone receptors such as *PPARα*, *GR* and *MR* are evident in both the RPE and choroid, while *PR* expression was lost in 1°CEC. *VDR* and *AR* were expressed in both models of CEC and overlap with 1°RPE but not fRPE. While *ERβ* and *PNR* were exclusively present in 1°CEC, and *ERRβ* and *FXR* were only detected in fCh. No receptors were found to be solely expressed in either 1°RPE or fRPE. *RORγ* was a common nuclear receptor expressed in 1°RPE and fRPE and absent/undetectable in the choroid models. Finally, *FXR*, which is closely related to the *LXRs* was only found to be expressed in fCh. Collectively, these outcomes reveal the multidimensional expression patterns of the different members of this transcription family in the complex RPE-choroid, critical to be aware of when investigating the impact of receptor agonism or antagonism in the eye and/or assessing therapeutic impact of targeting nuclear receptors.

### 3.5. Nuclear Receptors Are Differentially Regulated Following RPE/Choroidal Injury

The laser induced choroidal neovascularization (CNV) model is a pre-clinical model, in which a laser is used to create a bubble in the RPE layer that triggers a break through Bruch’s membrane to allow for new vessel growth from the outer choroid. Given age is an important factor in AMD, we created neovascular lesions in aged C57BL/6J mice, in order to identify the nuclear receptors involved in the development of CNV, the results of which were compared to a cohort of mice not subjected to laser damage. The injury and in vivo imaging paradigms are shown in a schematic in Figure 5A. Fundus, OCT and fluorescein angiography images confirm the formation of neovascular lesions with defined borders following laser treatment (Figure 5B–D) as compared to the healthy fundus with intact retinal layers seen in the control, non-lasered group (Figure 5A,H,I). The cellularity and vascular regions of the neovascular lesions were visualized, respectively, in propidium iodide and isolectin GS-IB4 stained RPE flat mounts compared to non-lasered controls (Figure 5A,F,J,K; lesions are delineated by a circle). Following isolation of the choroidal tissues from the non-lasered and lasered mouse eyes, the expression of nuclear receptors, coactivators, corepressors and chromatin modifying enzymes were determined using PCR arrays and clustergram analysis was performed on the Geneglobe analysis web portal (Figure 6). Differentially regulated genes were plotted on a heat map to demonstrate log 2 fold changes of upregulated and downregulated genes (Appendix A). Fold regulation values for all the 84 tested genes are presented in Figure 7A grouped based on ligand binding. Next, we plotted log10 gene expression changes of the lasered versus non-lasered groups presented as a scatter plot to highlight gene expression changes above a 2-fold difference threshold (Figure 7B). A total of 15 disease-associated candidate nuclear receptors and coregulators were discovered to be downregulated by 2-fold or more in the laser-induced CNV group as highlighted by yellow dots on the scatter plot (Figure 7B). *VDR* was the only upregulated nuclear receptor, though it fell below the 2-fold threshold in the experimental CNV group as compared to non-lasered group. Table 2 lists the 15 downregulated genes with a 2-fold or greater difference and the *p*-values for each nuclear receptor. Downregulated genes included members of the steroid hormone receptor family including *ERα*, *MR* and *PR*. Additionally, *RXRα* and *RXRβ*, were downregulated but not *RXRγ*. Other downregulated genes included *PPARα*, shown to play a role in retinopathy of prematurity [51], *REV-ERBα* and *REV-ERBβ*, potential regulators of circadian rhythm and receptors which have been linked to retinitis pigmentosa (RP) through regulation of *PNR* [52]. *NUR77*, *ERRα*, *ERRβ* and *ERRγ* were also all downregulated, with *ERRβ* and *ERRγ* showing 6.56 fold and 3-fold downregulation respectively. Finally, mediator complex subunit 16 (*MED16*), which is an important subunit of the multimeric protein complex associated with nuclear receptors and RNA polymerase II, along with the peroxisome proliferator-activated receptor gamma coactivator 1-alpha (*PPARGC1A*), which is an important coactivator of *PPARγ*, involved in regulating energy metabolism, were two co-regulators found to be downregulated in the neovascular choroid. Collectively, our studies have revealed candidate receptors and co-regulators that are associated with vascular disease of the posterior pole of the mouse eye.

## 4. Discussion

The choroid plays an important role in visual function as it supports the delivery of nutrients and facilitates the removal of waste products from the overlying RPE and neural retina, which houses the light sensitive photoreceptors. The significance of the choroid is further highlighted as its dysfunction and/or degeneration are involved in several ocular blinding diseases including age-related macular degeneration, choroidal dystrophies, diabetic retinopathy, choroidal retinopathy, and uveitis, among others. Herein we developed a nuclear receptor atlas of the choroid and choroidal endothelial cells. This entailed determining the expression profile of all members of this transcription family, in two model systems derived from human donor eyes: cultured choroidal endothelial cells and freshly isolated choroidal tissue. Our goals in creating this atlas were three-fold: (1) to catalog the distribution and expression levels of the receptors in the outer ocular vasculature tissues, (2) to determine the degree to which there is variability in expression between the single cell culture system of choroidal endothelial cells versus the more complex choroidal tissue, and (3) to identify candidate receptors involved in choroid associated disease processes. The overlap in expression between the two model systems would support utilizing primary cell cultures of choroidal endothelial cells, as a platform, for further investigating those common individual receptors under normal and disease conditions.

Broadly, we found significant overlap in the presence of nuclear receptors between the two model systems. There were, however, some exceptions, with select receptors detected in one but not the other sample group. The most notable example was the detection of *PNR* in our 1°CEC samples. *PNR* is highly enriched in the retina and expressed at much lower levels in other tissues throughout the body including the thyroid, prostate, and liver. Mutations in *PNR* have been associated with several forms of retinal degenerations, the receptor is considered to be a broad-spectrum genetic modifier [53], and in partnership with *REV-ERBα* has been shown to act synergistically in the retina [54]. *PNR* has also been reported to regulate *ERα* expression in breast cancer cells [55], to be a critical component of p53 activation, and a potential susceptibility factor for toxicant-induced injuries to the liver [56]. Though baseline expression of the *PNR* gene in the choroidal endothelial cell line was shown, its potential homeostatic role in these cells and/or involvement in choroid associated diseases remains to be discovered.

Cross-comparing the nuclear receptor expression profile of the RPE, previously published by our group, with our current study, allowed for a deeper dive into the potential baseline roles these transcription factors may play in RPE-choroid biology. Furthermore, these data in combination with the nuclear receptor and co-modulator expression profiles of the aged laser-induced experimental CNV mouse model highlighted a number of candidate receptors that may play a role in wet AMD. As expected, *AhR* and its receptor *ARNT*, which have been reported to regulate angiogenesis, inflammation and extracellular matrix regulation [36,57], were found to be present in both the RPE and choroidal model systems. *PPARα*, important in lipid metabolism and inflammation along with *PPARβ/δ1*, important in neovascularization, were also expressed in both the RPE and choroid [32,58,59]. The two liver X receptors isoforms, *LXRα* and *β*, were also detected in the RPE and choroid and are important in ocular biology and AMD pathogenesis as they can be activated by cholesterol, and regulate transcription of lipid transporters including ATP binding cassette family A, member 1 (*ABCA1*) and cholesterol esterase transfer protein (*CETP*). Importantly, they also modulate inflammation, a key pathogenic pathway in AMD development. GR and MR, traditionally known as corticoid receptors are two other receptors that may be relevant to choroidal homeostasis, as they are both expressed in various tissues in the eye, including the RPE and choroid, and have anti-inflammatory roles [60]. In a rabbit model of CNV, synthetic corticosteroid (triamcinolone acetonide) treatment decreased VEGF and inflammatory cytokines [61], while activation of the mineralocorticoid receptors are vital in maintaining the tone of vasculature (mainly endothelial cells and smooth muscle cells) [62,63] and antagonism of the receptor has been shown to improve choroidal vasculopathy and lessen the severity of laser-induced CNV in mice [64]. *VDR*, is another receptor worth highlighting, found to be expressed in 1°CECs and fCh. Importantly, its expression increased following laser injury in our CNV mouse cohort. Mechanistically, VDR has been reported to play regulatory roles during the development of the retinal vasculature, in pathological neovascularization [65] and is highly expressed in retinal pericytes as compared to endothelial cells [66]. Though high serum levels of vitamin D have been associated with potential protection in AMD [67,68], the role of VDR in choroidal pericytes and choroidal biology itself has not been explored in detail.

The *NR3B* family of nuclear receptors also known as estrogen-related receptors (*ERRs*) regulate metabolic genes involved in glucose and glutamine metabolism, playing an important role in energy metabolism. As such, they have been implicated in metabolic disorders such as type 2 diabetes and metabolic syndrome [69]. Overexpression of *ERR**γ* has been shown to increase the expression of matrix metalloproteinase (*MMP9*) and vascular endothelial growth factor A (*VEGFA*) in chondrocytes, resulting in extracellular matrix degradation and vascular proliferation in osteoarthritis. These pathways are also relevant to AMD [70]. The *ERRβ* isoform has been shown to regulate placental development and stem cell pluripotency in mice. Herein we noted that *ERRβ* and *NCOA3,* an essential coactivator required to mediate *ERRβ* function in embryonic stem cells [71], were downregulated in the choroid of CNV-lasered mice, supporting a potential mechanistic role in wet AMD. Since orphan nuclear receptors are able to bind to DNA not only as monomers, but also as homodimers or heterodimers [72,73], it is conceivable that *ERRβ* may work cooperatively with other receptors including *REV-ERBα* and *β* and/or *NUR77*, which can recognize extended half sites [74,75,76]. This speculative interplay of nuclear receptors should be further investigated, in order to clarify this potential transcriptional network, in particular given our observation that *ERRβ*, *REVERBα*, *REVERBβ* and *NUR77* are all downregulated in the experimental wet AMD mouse model.

The multi-functional estrogen receptors support a variety of biological processes throughout the body including reproduction, cardiovascular health, bone integrity, cognition, and behavior. In the eye, *ERs* regulate *MMP2* in RPE cells, relevant since dysregulation of MMPs has been implicated in AMD development [77,78,79]. Additionally, decreased estrogen levels in postmenopausal women have been presented as a potential reason for the higher incidence of AMD in women [80,81,82]. We found *ER* to be expressed in human choroidal endothelial cells and freshly isolated choroids (Figure 4). In discovering the potential role *ERs* play in the eye, it is important to note that like many other nuclear receptors, ER function is highly cell and tissue-specific. Specifically, while estrogen has been indicated to contribute positively through *ERα* activation to improved cardiovascular health [83,84,85], it has also been reported to facilitate cell proliferation in cancer [86]. Dichotomous roles for ER have been reported in bone tissue [87,88]. Finally, estrogen has been reported to exert a proangiogenic effect and participate in endothelial repair endowing beneficial effects mainly via ERα activation. In line with these observations, it is not surprising that angiogenesis is impaired in *ER* knockout mice [89,90]. In sum, in pathological circumstances, such as breast cancer, a clear association has been observed between estrogen levels, *ER* expression by endothelial cells, angiogenic activity and/or tumor invasiveness [91].

Nuclear receptors, which act as ligand inducible transcriptions factors, can activate or repress gene expression by directly binding to the DNA response elements of target genes. Transcriptional regulation by nuclear receptors is mediated by recruitment of coactivators or corepressors, effecting chromatin remodeling and histone modifications [92]. The peroxisome proliferator-activated receptor gamma coactivator 1-alpha (*PPARGC1A*), a nuclear receptor co-modulator, was downregulated in the choroidal tissues of the laser-induced CNV mouse, consistent with the observation that *PPARβ/δ* regulates signaling pathways involved in neovascular lesion development, thus potentially contributing to the pathogenesis of AMD [32]. Mechanistically, *PPARGC1A* has been reported to increase the expression of B-type natriuretic peptide (*BNP*) expression through co-activation of *ERRα* and/or *AP1* in muscle cells [93]. *BNP* is thought to aid in tissue repair and regeneration in muscles, providing a testable hypothesis on how the downregulation of *PPARGC1A* may affect tissue repair and regeneration following laser injury. *MED16*, is another important mediator complex subunit, shown to be required for transcription by RNA polymerase II. Together with *MED12*, *MED16* cooperates to suppress the activation of the TGFβ pathway. *MED12* expression has been reported to be downregulated in papillary thyroid cancer leading to activation of TGFβ signaling [94]. In the experimental laser-induced CNV mouse model, *MED16* was found to be downregulated, interesting, as the TGFβ pathway is active in CNV [95]. These findings support the targeting of *MED16* and/or the use of TGFβ inhibitors as potential therapeutics for managing wet AMD treatment.

There are a number of important observations worth elaborating on and considering in interpreting the data in our study. First, the primer/probe sets for the nuclear receptors in our study were validated for optimal PCR efficiency (90–110%) using a template titration assay of universal human cDNA and based on a single peak in the dissociation curve, slope, and R^2^ value (>0.95) of the standard curve plot of Ct value vs. cDNA [31]. Further confirmation of nuclear receptor levels at the protein level, on an individual basis, in future studies, is needed to complement the qPCR results. Second, the cohort used in the laser CNV study included a greater number of male mice aged 24–28 months versus females. Though there is no consensus on differences in longevity between male and female mice, in our colony more male mice survived and one may conclude that the nuclear receptor expression pattern presented here may be representative of males. However, an evaluation of a larger female cohort would be necessary, in order to definitively identify potential sex-dependent differences in the expression of nuclear receptors in the choroidal tissues. Third, there are demonstrated regional differences in the morphology and gene expression of the macular versus peripheral compartments of the eye [96,97], significant as the macula is particularly vulnerable to diseases such as AMD. It follows, that it is plausible there may be differences in nuclear receptor expression levels in the macula and periphery, which will need to be explored in detail during follow-up and focused investigation of individual candidate receptors in human ocular tissue samples. Finally, the freshly isolated choroid data should not be directly compared to that of the single population of choroidal endothelial cells. Both the human and mouse choroidal tissues analyzed have a rich composition of melanocytes, fibroblasts, immune cells, smooth muscle cells, supporting collagenous and elastic connective tissue, and pericytes. Therefore, the single cell population versus complex tissue should be considered when evaluating, interpreting the data, and exploring potential therapeutics.

## 5. Conclusions

This is the first report of three comprehensive nuclear receptor atlases of human choroidal endothelial cells, human choroidal tissue, and mouse choroid tissue following laser-induced CNV induction. Overall, these nuclear receptor atlases, in combination with the previously published RPE atlas, are useful resources and frameworks to further study the role of nuclear receptors in ocular biology and disease. Importantly, they highlight the eye as a secondary endocrine organ, in which nuclear receptors play an important role not only in normal cellular homeostasis but also disease. Given the availability of FDA-approved drugs for several of these receptors, there is potential for expedited bench-to-bedside application for select receptors should, upon further investigation, be confirmed to have therapeutic potential.

## Figures and Tables

**Figure 1 cells-11-02386-f001:**
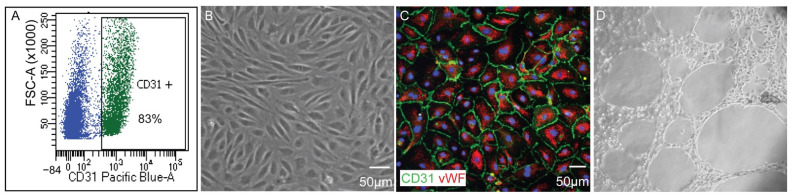
Characterization of human primary choroidal endothelial cells (1°CEC). (**A**) 1°CEC stain 83% positive for CD31 as demonstrated by flow cell cytometry analysis. (**B**) 1°CEC grown to confluence maintain a “cobblestone-like” appearance as observed with light microscopy. (**C**) 1°CEC stain positive for pan-endothelial surface marker CD31 (green) and Von Willebrand Factor (vWF, red). (**D**) Tube formation assay of 1°CECs. FSC-A = forward scatter area; aquablue = live-dead stain for live cell gating. Magnification bar = 50 μm.

**Figure 2 cells-11-02386-f002:**
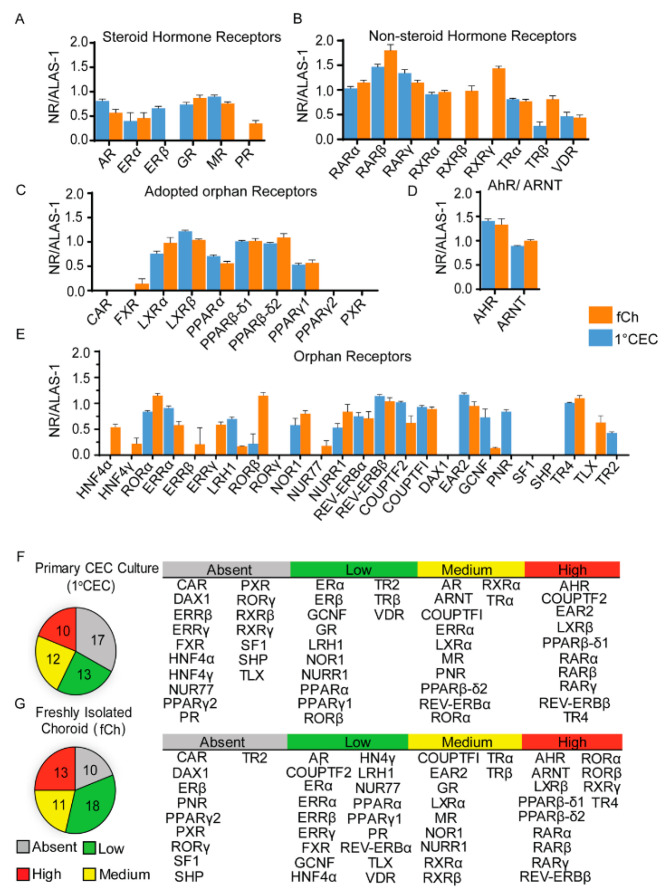
Nuclear receptor expression profiles by qPCR analysis of primary human choroidal endothelial cells (1°CEC) and freshly isolated human choroid (fCh). Expression profiles of nuclear receptors and related transcription factors are categorized into steroid hormone nuclear receptors (**A**), non-steroid hormone receptors (**B**), adopted orphan nuclear receptors (**C**), AhR and ARNT (**D**), which are not classical nuclear receptors but important transcription factors involved in toxin clearance and orphan nuclear receptors (**E**). Data for nuclear receptors are presented as mean arbitrary expression ratios ± SEM for 1°CEC and fCh. The levels of nuclear receptor expression in 1°CEC (**F**) and fCh (**G**) are displayed in pie charts on the left, and the nuclear receptor common names are listed according to their detectable levels on the right. Normalized nuclear receptor mRNA expression levels were defined as absent if the Ct value was ≥35, low if the level was less than 0.75, medium if the level was between 0.75 and 1.0, and high if the level was greater than 1. NR = nuclear receptor.

**Figure 3 cells-11-02386-f003:**
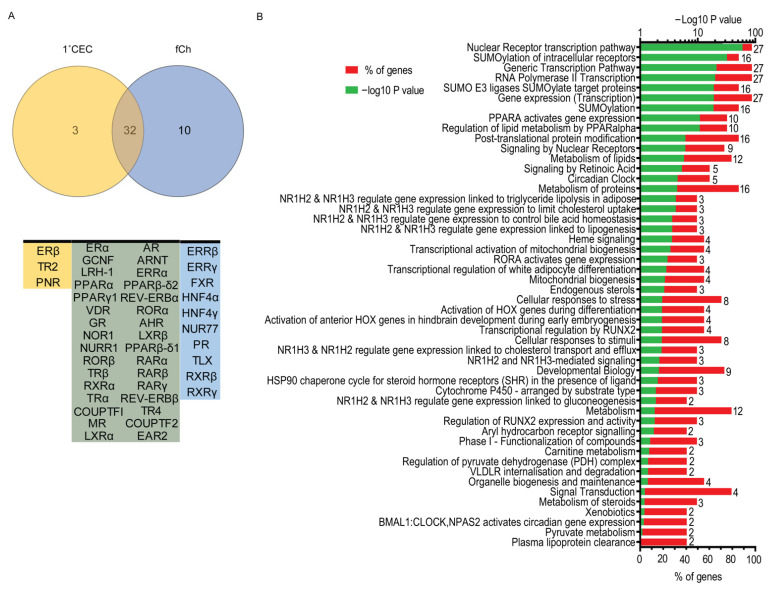
Nuclear receptor expression in human choroidal endothelial cells compared to the choroid and biological pathway analyses. (**A**) Venn diagram illustrates overlapping expression of nuclear receptors in primary choroidal endothelial cells (1°CEC) and freshly isolated choroidal endothelial cells (fCh). Diagram was made using Venny 2.1.0. (**B**) Gene Ontology analysis of 32 common nuclear receptors in the 1°CEC and fCh were significantly enriched in a number of pathways. Stacked bar graph shows % of genes in the pathway (red), −log10 *p* value of each category in green and number of genes next to each bar graph. Analysis was performed using the DAVID functional annotational tool with the Reactome database.

**Figure 4 cells-11-02386-f004:**
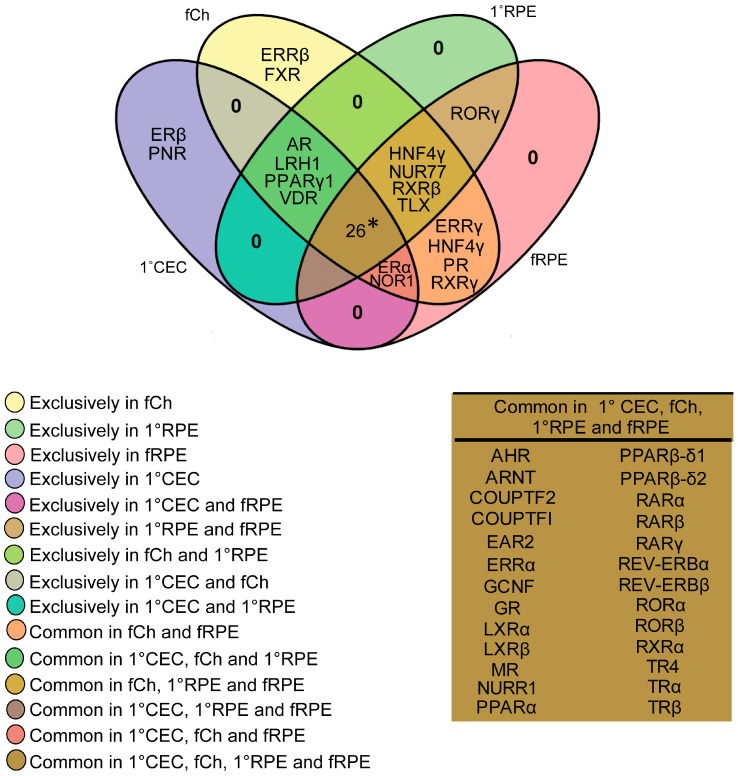
Commonly expressed nuclear receptors in the human choroidal tissue models versus RPE models. Commonly expressed nuclear receptors are shown in a four-set Venn diagram, with an overlap of freshly isolated choroid tissue (fCh), cultured primary choroidal endothelial cells (1°CEC), freshly isolated retinal pigment epithelial cells (fRPE) and cultured primary RPE is presented here. The Venn diagram was drawn using VENNY 2.1.0. Common nuclear receptors expressed in all four categories are listed in the table with color coded legend. * 26 common nuclear receptors found in all model systems are shown in the adjoining table.

**Figure 5 cells-11-02386-f005:**
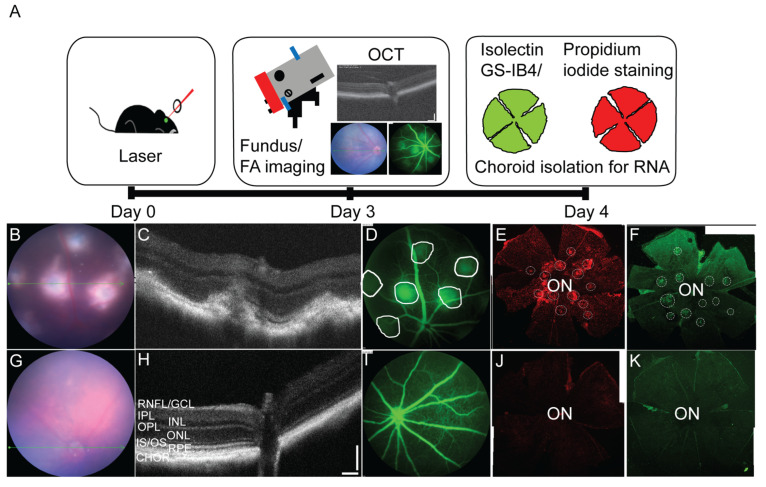
Laser induced CNV mouse disease model paradigm. (**A**) Schematic of the experimental plan. On day 0 lesions in mice eyes are induced with laser and imaged at day 3 with optical coherence tomography (OCT), fundus and fluorescein angiography. On day 4, eye tissues are collected for flatmount staining with isolectin GS-IB4 and propidium iodide and choroid is isolated for RNA extraction. In vivo evaluation of mice 3 days post lasering occurred in 15–17 month mice. Representative images are shown here. (**B**,**G**) Fundus images of the posterior eye showing the extent of the CNV lesions. (**C**,**H**) OCT images displaying cross-sections of the lesions (corresponding to the green line in ‘(**B**)’ or ‘(**G**)’). (**D**,**I**) Regions of leakage (dotted line circles) from the CNV lesions are visible in fluorescein angiography images. (**E**,**J**). A montage of the entire choroidal flat mount was created by overlapping and combing 4–5 images captured at lower magnification (4×). Choroidal flat mounts from control and CNV lasered mice stained with propidium iodide and (**F**,**K**) isolectin GS-IB4Representative image is shown to demonstrate CNV lesions (dotted line circles demarcate lesions). CNV: Choroidal neovascularization; ON: Optic nerve head; RNFL/GCL: retinal nerve fiber layer/ganglion cell layer; IPL: inner plexiform layer; INL: inner nuclear layer; OPL: outer plexiform layer; ONL: outer nuclear layer; IS/OS: inner segment/outer segment; RPE: retinal pigment epithelium; Chor: choroid.

**Figure 6 cells-11-02386-f006:**
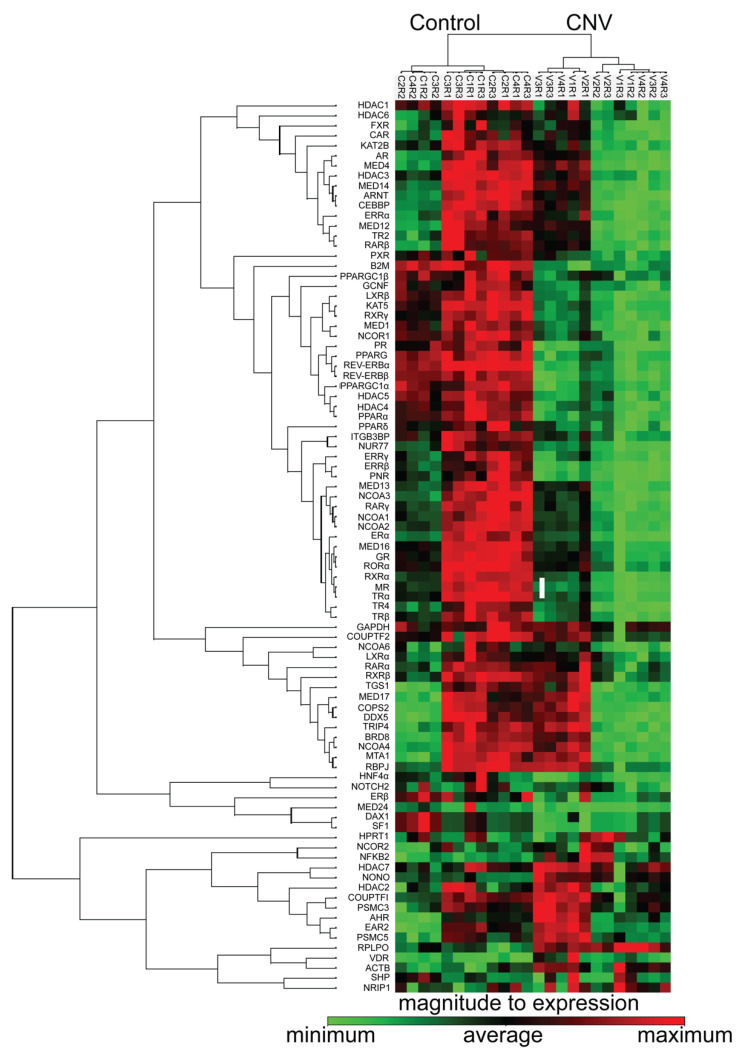
Heatmap of nuclear receptor expression in the choroid of aged laser induced CNV mice. Clustergram shows expression intensity results of cluster analysis of 84 genes from the PCR array in the control and laser induced CNV mice cohorts (*n* = 12 arrays per group). Each colored band represents expression of a single gene from a sample, with higher expression in shown in red and lower expression shown in green.

**Figure 7 cells-11-02386-f007:**
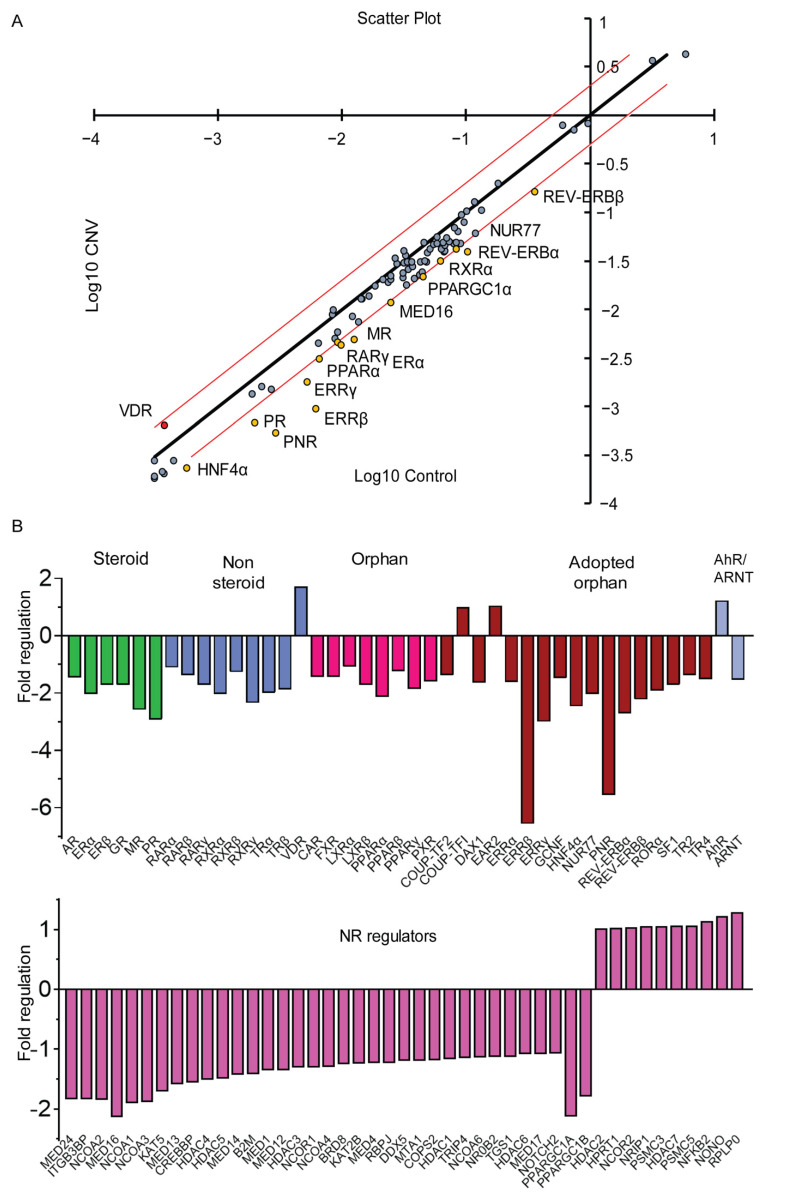
Nuclear receptor expression in the choroid of age laser induced CNV mice. (**A**) Scatter plot showing gene expression changes of nuclear receptors and co-regulators in lasered mice as compared to control group. The scatter plot compares the normalized expression of every gene on the PCR array between the two selected groups by plotting them against one another to visualize large gene expression changes. The center diagonal line indicates unchanged gene expression, while genes with red data points above in the upper left and yellow data points in lower right corners are up-regulated or down-regulated, respectively by more than 2-fold regulation threshold in the y-axis CNV Group relative to the x-axis Control Group. Grey data points represent within 2-fold cutoff threshold indicated by red lines. (**B**) Fold regulation of genes in CNV laser versus control mice using ΔΔCT method. Fold regulation of genes are categorized into different nuclear receptor classes like steroid hormone receptor, non-steroid hormone receptor, orphan receptors and adopted orphan receptors. NR = nuclear receptor.

**Table 1 cells-11-02386-t001:** Differentially expressed genes in primary CECs and freshly isolated choroid. Differentially expressed genes in two models of human choroidal endothelium are presented. Normalized nuclear receptor mRNA expression levels were defined as absent if the Ct value was ≥35, low if the level was less than 0.75, medium if the level was between 0.75 and 1.0, and high if the level was greater than 1. Absent (−) Low (+) Medium (++) High (+++).

Gene	1°CEC	fCh
*AR*	(++)	(+)
*ARNT*	(++)	(+++)
*COUP-TF2*	(+++)	(+)
*EAR2*	(+++)	(++)
*ERRα*	(++)	(+)
*ERRβ*	(−)	(+)
*ERRγ*	(−)	(+)
*ERβ*	(+)	(−)
*FXR*	(−)	(+)
*GR*	(+)	(++)
*HNF4α*	(−)	(+)
*HNF4γ*	(−)	(+)
*NUR77*	(−)	(+)
*NOR1*	(+)	(++)
*NURR1*	(+)	(++)
*PNR*	(++)	(−)
*PPAR-β/δ2*	(++)	(+++)
*PR*	(−)	(+)
*REV-ERBα*	(++)	(+)
*RORα*	(++)	(+++)
*RORβ*	(+)	(+++)
*RXRβ*	(−)	(++)
*RORγ*	(−)	(+++)
*TLX*	(−)	(+)
*TR2*	(+)	(−)
*TRβ*	(+)	(++)

**Table 2 cells-11-02386-t002:** List of downregulated genes in choroids of the experimental laser induced CNV mouse model. CECs were isolated from mice subjected to laser injury and control group, not treated with laser. Nuclear receptor and co-regulator expressions were evaluated by PCR arrays. Analyses were performed on the Geneglobe analysis portal.

Sr. No.	Gene	Alias	Fold Regulation	*p*-Value
1.	*ERα*	Estrogen Receptor 1	−2.04	0.000066
2.	*ERRβ*	Estrogen Related Receptor Beta	−6.56	0
3.	*ERRγ*	Estrogen Related Receptor Gamma	−3	0.000009
4.	*HNF4α*	Hepatocyte Nuclear Factor 4 Alpha	−2.46	0.000059
5.	*MED16*	Mediator Complex Subunit 16	−2.14	0.000001
6.	*REV-ERBα*	V-ErbA-Related Protein 1	−2.71	0
7.	*REV-ERBβ*	V-ErbA-Related Protein 1-Related	−2.22	0
8.	*PNR*	Photoreceptor-Specific Nuclear Receptor	−5.55	0
9.	*MR*	Mineralocorticoid Receptor	−2.58	0
10.	*NUR77*	Nuclear Hormone Receptor NUR/77	−2.03	0.000002
11.	*PR*	Progesterone Receptor	−2.93	0
12.	*PPARα*	Peroxisome Proliferator Activated Receptor Alpha	−2.14	0
13.	*PPARGC1A*	PPARG Coactivator 1 Alpha	−2.13	0
14.	*RXRα*	Retinoid X Receptor Alpha	−2.03	0.000004
15.	*RARγ*	Retinoid X Receptor Gamma	−2.34	0

## Data Availability

All data relevant to the study are included in the article or uploaded as Appendix A.

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
