# Peer review of "Nuclear Receptor Atlases of Choroidal Tissues Reveal Candidate Receptors Associated with Age-Related Macular Degeneration"

_cells, 2022, doi:10.3390/cells11152386_

Round 1
Reviewer 1 Report
The work by Peavey and collaborators nicely provides data regarding the expression of nuclear receptors in choroidal endothelial cells. These data may be a useful resource to establish the influence of nuclear receptor signaling pathways on the outer vascular tissue of the eye. The work is nicely designed and performed.
Minor details:
-Why did authors use many more male than female mice in their study?
- Figure 1D is not properly visualized
_Figure 2. A legend describing what are the blue and orange bars in the graphs is missing.
Reviewer 2 Report
This manuscript is well written and presents interesting data on the expression of nuclear receptors in choroidal endothelial cells that is validated by freshly isolated choroid tissue from human donor eyes. This reviewer commends the work of these authors as it will be an important resource for the AMD research community, especially those interested in choroidal biology and identifying nuclear receptor candidates in the choroid for more in-depth analyses in the future.
There are few questions and considerations from this reviewer for the authors of this work:
1) Is the eye disease status known for the eyes used in this study? If it is not known, then it may be worthwhile to note this in the discussion as it may effect the expression of nuclear receptors in the choroid.
2) Does the authors postulate that there are differences in nuclear receptor gene expression between the choroidal endothelial cells in the periphery and macula regions of the retina? This may be worth commenting on in the discussion of this manuscript.
3) The authors include a nice paragraph at the end of their discussion about the limitations of freshly isolated choroid tissue from human donor eyes. However, the same could be said about the freshly isolated mouse choroid. This tissue includes immune cells, fibroblasts, and other cells. There should be discussion about the limitations of the laser-induced CNV model as well as careful interpretation of the results from this study. This reviewer also suggests changing references of mouse choroid to mouse choroidal tissue as it is stated for the human choroidal tissue isolated from human donor eyes.
Comments on Figures:
1) There should be a figure legend for Figure 2A-E that shows blue for primary choroidal endothelial cells and orange for freshly isolated choroid.
2) This reviewer suggests Figure 3A-B and Figure 3C be included into separated figures. This may allow the text in Figure 3B to be enlarged since it is quite small.
3)Figure 5 is quite busy and a lot of the text is quite small. This reviewer wonders if all the panels of the figure are necessary and suggests the authors revise it. At the very least, the text in the figure needs to be enlarged in order to read it.
Author Response
We thank the reviewers for their time reading our manuscript and provided constructive feedback. We address the concerned raised and provide our point by point responses below.
We thank the journal for considering our study for publication and looking forward to hearing from you.
Sincerely,
-Goldis Malek, PhD
Reviewer 2:
This manuscript is well written and presents interesting data on the expression of nuclear receptors in choroidal endothelial cells that is validated by freshly isolated choroid tissue from human donor eyes. This reviewer commends the work of these authors as it will be an important resource for the AMD research community, especially those interested in choroidal biology and identifying nuclear receptor candidates in the choroid for more in-depth analyses in the future.
There are few questions and considerations from this reviewer for the authors of this work:
- Is the eye disease status known for the eyes used in this study? If it is not known, then it may be worthwhile to note this in the discussion as it may effect the expression of nuclear receptors in the choroid.
Donor eyes obtained were reported to not have a past history of ocular disease. We have added this important information to the Materials and Methods section, 2.1. Model Systems on page 4, lines 96-97.
- Does the authors postulate that there are differences in nuclear receptor gene expression between the choroidal endothelial cells in the periphery and macula regions of the retina? This may be worth commenting on in the discussion of this manuscript.
This is a great point and though we did not look for regional differences, there are a number of studies which have elegantly teased out differential expression in the macular versus the peripheral retina and RPE/choroidal tissue. We have added a statement regarding this to the last paragraph of the manuscript on page 17, lines 515-519.
- The authors include a nice paragraph at the end of their discussion about the limitations of freshly isolated choroid tissue from human donor eyes. However, the same could be said about the freshly isolated mouse choroid. This tissue includes immune cells, fibroblasts, and other cells. There should be discussion about the limitations of the laser-induced CNV model as well as careful interpretation of the results from this study. This reviewer also suggests changing references of mouse choroid to mouse choroidal tissue as it is stated for the human choroidal tissue isolated from human donor eyes.
Thank you for your recommendation. We agree and have updated the text to refer to the mouse choroid as “mouse choroidal tissue” to be consistent with the nomenclature we used for the human tissue samples. We have also updated the title and replaced ‘choroid’ with ‘choroidal tissues’. Finally, we have expanded the last paragraph to include noteworthy observations for the reader to consider when interpreting the nuclear receptor atlases.
Comments on Figures:
- There should be a figure legend for Figure 2A-E that shows blue for primary choroidal endothelial cells and orange for freshly isolated choroid.
Thank you for catching that oversight. We have added the legend.
- This reviewer suggests Figure 3A-B and Figure 3C be included into separated figures. This may allow the text in Figure 3B to be enlarged since it is quite small.
Agreed. Figure 3C is new Figure 4 and the panels have been enlarged.
3) Figure 5 is quite busy and a lot of the text is quite small. This reviewer wonders if all the panels of the figure are necessary and suggests the authors revise it. At the very least, the text in the figure needs to be enlarged in order to read it.
Agreed. We have moved 5B to the supplementary section and split the figure into a new figure 5 (includes previous 5A panel – enlarged) and new figure 6 (includes previous 5C and 5D panels).
Reviewer 3 Report
This manuscript studied the profile of nuclear receptors in the choroidal endothelial cells and they also studied the changes in the expression levels of nuclear receptors in a laser induced choroidal injury mice model.
The manuscript provides useful information to clearly demonstrate the nuclear receptor expression profiles in both RPE and choroidal endothelial cells. However, this manuscript did not offer the potential signaling pathways associated with AMD. In Figure 5, it only shows differential expression levels of genes. The authors should modify the title of this manuscript.
The most confusing part for me is that the authors use real-time PCR to confirm the levels of nuclear receptors. Why? An RNA-seq or single cell RNA-seq experiment would provided more and much reliable expression of those nuclear receptors. There are plenty bias in the real-time PCR experiments especially when they are using SYBR Green system. The potential of non-specific amplification of DNAs would provide biased data. In addition, it is highly recommended to include western blot or immunohistochemistry/immunofluorescence staining to affirm the levels of nuclear receptors. The presence of mRNA does not mean that there are proteins expressed in the cell.
Please check the correctness of Figure 2D and E. I think they are reversed.
Author Response
We thank the reviewers for their time reading our manuscript and provided constructive feedback. We address the concerned raised and provide our point by point responses below.
We thank the journal for considering our study for publication and looking forward to hearing from you.
Sincerely,
-Goldis Malek, PhD
Reviewer 3:
This manuscript studied the profile of nuclear receptors in the choroidal endothelial cells and they also studied the changes in the expression levels of nuclear receptors in a laser induced choroidal injury mice model.
The manuscript provides useful information to clearly demonstrate the nuclear receptor expression profiles in both RPE and choroidal endothelial cells. However, this manuscript did not offer the potential signaling pathways associated with AMD. In Figure 5, it only shows differential expression levels of genes. The authors should modify the title of this manuscript.
The nuclear receptor atlases are databases of candidate receptors that may play an important role in choroidal homeostasis and potentially choroidal neovascularization, the mechanisms of each of which would have to be explored in greater detail. We have considered this recommendation and have modified the title to:
Nuclear receptor atlases of choroidal endothelial cells and choroidal tissues reveal candidate receptors associated with age-related macular degeneration
The most confusing part for me is that the authors use real-time PCR to confirm the levels of nuclear receptors. Why? An RNA-seq or single cell RNA-seq experiment would provided more and much reliable expression of those nuclear receptors. There are plenty bias in the real-time PCR experiments especially when they are using SYBR Green system. The potential of non-specific amplification of DNAs would provide biased data. In addition, it is highly recommended to include western blot or immunohistochemistry/immunofluorescence staining to affirm the levels of nuclear receptors. The presence of mRNA does not mean that there are proteins expressed in the cell.
We appreciate this comment. The intent of our study was to interrogate the expression levels of a family of related genes in a purposeful manner. qPCR and PCR arrays have been used by us and others to successfully create comprehensive nuclear receptor atlases of tissues throughout the body in the past, data from which has led to focused studies on candidate receptors as a function of development, age, and disease [1-4]. There are a number of excellent RNA-seq studies in the literature that have cataloged all transcripts, regardless of their structure or function, born out of a different intent. This was not our goal.
Our future studies will include detailed mechanistic investigations on the role of select receptors in choroidal tissues as a function of age and disease. In a manner similar to the mechanistic and focused studies using molecular biology, histological, immunohistological studies in vitro and in vivo on select receptors pursued by us following the development of the RPE nuclear receptor atlas previously [5-9]. As there are a total of 48 nuclear receptor and even more co-regulators, we respectfully submit that performing Western blot and/or IHC experiments on such a large scale is beyond the scope of the current paper.
Finally, we appreciate the limitations of qPCR, however, we validated our primers, which were selected from the NURSA (Nuclear Receptor Signaling Atlas) website and Harvard Primer bank, based on a single peak in the dissociation curve, slope, and R2 value (>0.95) of the standard curve plot of ct value vs. cDNA quantity as previously reported [3]. Briefly, the nuclear receptor specific primers were validated for optimal PCR efficiency using a template titration assay of universal human tissue cDNA representing a broad range of expressed genes. Two fold serial dilution of cDNA spanning 12 ng through 0.0002 ng of the reference cDNA were used as template for the standard curves to measure PCR efficiency of each NR qPCR primer set. That said, we appreciate the comments of the reviewer and have briefly stated limitations of qPCR in the final paragraph of the discussion on pages 16-17, lines 504-509.
Please check the correctness of Figure 2D and E. I think they are reversed.
Thank you. This has been corrected in the text.
References:
- Becnel, L.B., et al., Nuclear Receptor Signaling Atlas: Opening Access to the Biology of Nuclear Receptor Signaling Pathways. PLoS One, 2015. 10(9): p. e0135615.
- De Vitto, H., A.M. Bode, and Z. Dong, The PGC-1/ERR network and its role in precision oncology. NPJ Precis Oncol, 2019. 3: p. 9.
- Dwyer, M.A., et al., Research resource: nuclear receptor atlas of human retinal pigment epithelial cells: potential relevance to age-related macular degeneration. Mol Endocrinol, 2011. 25(2): p. 360-72.
- Li, Z., et al., Nuclear receptor atlas of female mouse liver parenchymal, endothelial, and Kupffer cells. Physiol Genomics, 2013. 45(7): p. 268-75.
- Choudhary, M., et al., PPARbeta/delta selectively regulates phenotypic features of age-related macular degeneration. Aging (Albany NY), 2016. 8(9): p. 1952-1978.
- Choudhary, M., et al., LXRs regulate features of age-related macular degeneration and may be a potential therapeutic target. JCI Insight, 2020. 5(1).
- Choudhary, M., et al., Aryl hydrocarbon receptor knock-out exacerbates choroidal neovascularization via multiple pathogenic pathways. J Pathol, 2015. 235(1): p. 101-12.
- Choudhary, M., S. Safe, and G. Malek, Suppression of aberrant choroidal neovascularization through activation of the aryl hydrocarbon receptor. Biochim Biophys Acta Mol Basis Dis, 2018. 1864(5 Pt A): p. 1583-1595.
- Hu, P., et al., Aryl hydrocarbon receptor deficiency causes dysregulated cellular matrix metabolism and age-related macular degeneration-like pathology. Proc Natl Acad Sci U S A, 2013. 110(43): p. E4069-78.
Round 2
Reviewer 3 Report
I still recommend the need to include protein expression levels rather than only gene levels.
Author Response
Thank you for your recommendation. To provide protein expression to over 48 nuclear receptors and then co-regulators is cost prohibited for us and beyond the scope of this study.